# A Modification of the PBIL Algorithm Inspired by the CMA-ES Algorithm in Discrete Knapsack Problem

**Maria Konieczka, Alicja Poturała, Jarosław Arabas** and **Stanisław Kozdrowski ***

Institute of Computer Science, Warsaw University of Technology, Nowowiejska 15/19, 00-665 Warsaw, Poland; maria.konieczka.stud@pw.edu.pl (M.K.); alicja.poturala.stud@pw.edu.pl (A.P.); jarabas@elka.pw.edu.pl (J.A.)
* Correspondence: s.kozdrowski@elka.pw.edu.pl

**Abstract:** The subject of this paper is the comparison of two algorithms belonging to the class of evolutionary algorithms. The first one is the well-known Population-Based Incremental Learning (PBIL) algorithm, while the second one, proposed by us, is a modification of it and based on the Covariance Matrix Adaptation Evolution Strategy (CMA-ES) algorithm. In the proposed Covariance Matrix Adaptation Population-Based Incremental Learning (CMA-PBIL) algorithm, the probability distribution of population is described by two parameters: the covariance matrix and the probability vector. The comparison of algorithms was performed in the discrete domain of the solution space, where we used the well-known knapsack problem in a variety of data correlations. The results obtained show that the proposed CMA-PBIL algorithm can perform better than standard PBIL in some cases. Therefore, the proposed algorithm can be a reasonable alternative to the PBIL algorithm in the discrete space domain.

**Keywords:** population-based incremental learning; covariance matrix adaptation evolution strategy; covariance matrix adaptation population-based incremental learning; knapsack problem; data correlation

## 1. Introduction

Evolutionary algorithms (EAs) are a valuable tool for solving many multidimensional and NP-hard [1] practical problems, mainly because they outperform traditional methods, whose high space complexity often disqualifies them from being used to solve complex problems. EAs are a large and diverse family of algorithms that find solutions to problems in continuous and discrete domains. Among the prominent classes of EAs, the Estimation of Distribution Algorithms (EDAs) [2] can be listed, in which the probabilistic model mechanism realizes the evolution mechanism.

One of the more straightforward and well-known EDAs initially used to solve discrete problems is the Population-Based Incremental Learning (PBIL) algorithm, first proposed by Baluja in [3]. This algorithm owes its simplicity to the fact that the probability distribution of the subsequent bits in the chromosome is independent, so both the point generation and the learning process can be performed separately for each variable. For these reasons, PBIL is popular and various modifications of the algorithm have been developed, introducing, among others, the probability vector multiplication [4,5], elite strategy [6] and non-parametric approach [7].

A different approach than EDAs in the EAs family is evolutionary strategies (ES). Their leading representative is the Covariance Matrix Adaptation Evolution Strategy (CMA-ES) algorithm proposed in [8]. Initially, CMA-ES was invented to find solutions to continuous domain problems, but later, modifications dedicated to discrete problems were developed [9]. Whereas CMA-ES, such as EDAs, uses a probability distribution to describe the population—a multivariate normal distribution—, they differ in other aspects. The key among them is how the parameters are updated—in EDAs, learning considers a set of

points, while CMA-ES considers a set of steps [10]. CMA-ES outperforms the EDA family of algorithms in many problems [10].

In this paper, we propose a modification of the PBIL algorithm, which, inspired by CMA-ES, introduces information about correlations between variables into the probability distribution model. In some actual applications, taking the correlations between decision variables into account can bring real benefits, e.g., a faster algorithm convergence. For instance, one can imagine a given problem with two global optima with significantly different realizations. By adding a covariance matrix, the algorithm can detect such a situation and distinguish between these solutions. Knapsack problems with a varying item parameter correlation [11] were chosen to compare the modified and standard PBIL algorithms. The selected test case is a well-known NP-hard optimization problem commonly applied to compare algorithms solving discrete tasks.

The rest of the paper is organized as follows: In Section 2, we detail investigated PBILs, including our proposed modification. Next, in Section 3, we present the test environment consisting of knapsack problems. Then, in Section 4, results and a discussion are provided. Finally, Section 5 gives out a summary of the research findings.

## 2. PBIL Algorithm

In algorithms from the family of EDAs, each population is represented by a probability distribution. In the case of PBIL [3], this is a vector consisting of the probabilities of obtaining a value of one at each position in the chromosome (1).

$$\mathbf{p}^t = [p_1^t, p_2^t, \ldots, p_n^t] \tag{1}$$

In each iteration, a population of M individuals was generated based on the probability distribution using the *sample*. The solutions were, then, evaluated according to the adopted objective function, and the N best solutions were selected (*select*). Based on the obtained $O^t$ subset, a vector of probabilities was updated in the *update* function. The algorithm ran in a loop until the assumed stopping condition was not met. The code is presented in Algorithm 1.

---

**Algorithm 1:** PBIL

1: $initialize(\mathbf{p}^0)$
2: $t \leftarrow 0$
3: **while** $!stop$ **do**
4:     $P^t \leftarrow sample(\mathbf{p}^t, M)$
5:     $O^t \leftarrow select(P^t, N)$
6:     $\mathbf{p}^{t+1} \leftarrow update(O^t, \mathbf{p}^t, a)$
7:     $mutate(\mathbf{p}^{t+1})$
8:     $t \leftarrow t + 1$
9: **end while**

---

The method of updating the probability vector was described in Formula (2), where $a$ is a learning rate and $\mathbf{x}$ is a binary vector representing a single solution to the problem. In each iteration of the algorithm, the vector is modified according to the frequency of ones on each gene in the set of $N$ best solutions.

$$\mathbf{p}^{t+1} = (1 - a) \cdot \mathbf{p}^t + a \cdot \frac{1}{N} \sum_{\mathbf{x} \in O^t} \mathbf{x} \tag{2}$$

The stop condition could be customized for the problem under study. Typically, a maximum number of iterations ($max_{iter}$) is specified. Additionally, the algorithm is assumed to stop when the probability vector stabilizes, i.e., if all vector elements are in a fixed neighbourhood of zero or unity ($\epsilon$).

*Proposed Modifications*

The algorithm introduced in [3] assumes the independence of the individual genes in the chromosome. Inspired by the CMA-ES algorithm described in [8], a Covariance Matrix Adaptation Population-Based Incremental Learning (CMA-PBIL) algorithm that attempts to account for correlations between variables was proposed.

In the CMA-PBIL algorithm, the probability distribution of population was described by two parameters: the covariance matrix ($\mathbf{C}^t$) and the probability vector ($\mathbf{p}^t$). The CMA-PBIL pseudocode is presented in Algorithm 2. The steps of the modified algorithm correspond to its original version. The changes only relate to how the points are generated and how the probability distribution parameters are updated.

---

**Algorithm 2:** CMA-PBIL

---

1: $initialize(\mathbf{p}^0, \mathbf{C}^0)$
2: $t \leftarrow 0$
3: **while** !$stop$ **do**
4:　　$P^t \leftarrow sample(\mathbf{p}^t, \mathbf{C}^t, M)$
5:　　$O^t \leftarrow select(P^t, N)$
6:　　$(\mathbf{p}^{t+1}, \mathbf{C}^{t+1}) \leftarrow update(O^t, \mathbf{p}^t, \mathbf{C}^t, a)$
7:　　$t \leftarrow t + 1$
8: **end while**

---

The procedure for updating the covariance matrix is outlined in Formula (3). It employs the same learning factor $a$ as during the modification of the vector $\mathbf{p}^t$. The coefficient $a$ was squared as the variance was the second central moment.

$$\mathbf{C}^{t+1} = (1 - a^2) \cdot \mathbf{C}^t + a^2 \cdot \mathbf{C}' \tag{3}$$

The matrix $\mathbf{C}'$ was computed at each iteration according to Formula (4), where the vector $\mathbf{s}^t$ consists of the frequencies of ones on each bit in the set $O^t$.

$$\mathbf{C}'(i, j) = \frac{1}{N} \sum_{\mathbf{x} \in O^t} (x_i - s_i^t)(x_j - s_j^t) \tag{4}$$

The method of updating the vector $\mathbf{p}^t$ did not change.

## 3. Experiments

As a test case, multiple variants of knapsacks problems were chosen. The task aimed to select from a set of $n$ items a subset that maximizes the overall value of the knapsack within the capacity limit (*Capacity*). Each item $i$ was described by two parameters—weight ($weight_i$) and value ($value_i$). The decision of whether the item was included or not in the knapsack was represented by variable $x_i \in \{0, 1\}$. The main task is defined in Equation (5), while the limit is defined in Formula (6).

$$\arg\max_{\mathbf{x}} \sum_{i=1}^{n} x_i \cdot value_i \tag{5}$$

$$\sum_{i=1}^{n} x_i \cdot weight_i \leqslant Capacity \tag{6}$$

The fitness was evaluated using Formula (7), where $P$ is the penalty coefficient. The objective function considered both the goals, which was to maximize the values of the items and the constraint.

$$f(\mathbf{x}) = \begin{cases} \sum_{i=1}^{n} x_i \cdot value_i, & \text{if} \quad \sum_{i=1}^{n} x_i \cdot weight_i \leqslant Capacity \\ \sum_{i=1}^{n} x_i \cdot value_i - P \cdot (\sum_{i=1}^{n} x_i \cdot value_i - Capacity), & \text{otherwise} \end{cases} \tag{7}$$

The problem was investigated on six different knapsack problems generated as proposed in [11,12]. Problems with different correlation intensities (uncorrelated, moderately correlated, and strongly correlated) and with different capacities were examined. As reported in [11], the higher the correlation, the more challenging the knapsack problem was expected to become. The sets consisted of $n = 100$ items with a maximum weight $v = 10$ and maximum offset $r = 5$. The method by which the weights, values and capacities were generated sequentially for the following test cases is shown in Table 1. The $\mathcal{U}_{[a,b]}$ represents the continuous uniform distribution within the range of $[a, b]$.

**Table 1.** Test cases of knapsack problem.

| No. | Weight | Value | Capacity |
|:---:|:---:|:---:|:---:|
| 1 | $\mathcal{U}_{[1,v]}$ | $\mathcal{U}_{[1,v]}$ | $2 \cdot v$ |
| 2 | $\mathcal{U}_{[1,v]}$ | $\mathcal{U}_{[1,v]}$ | $\frac{1}{2} \sum_{i=1}^{n} weight_i$ |
| 3 | $\mathcal{U}_{[1,v]}$ | $weight_i + \mathcal{U}_{[-r,r]}$ | $2 \cdot v$ |
| 4 | $\mathcal{U}_{[1,v]}$ | $weight_i + \mathcal{U}_{[-r,r]}$ | $\frac{1}{2} \sum_{i=1}^{n} weight_i$ |
| 5 | $\mathcal{U}_{[1,v]}$ | $weight_i + r$ | $2 \cdot v$ |
| 6 | $\mathcal{U}_{[1,v]}$ | $weight_i + r$ | $\frac{1}{2} \sum_{i=1}^{n} weight_i$ |

The initialization method for vector **p** and matrix **C** was adapted to the test case. Typically, in the PBIL algorithm, the computation starts with a vector $\mathbf{p}^0 = [0,5 \ldots 0,5]$. For the experiments performed, the elements of the vector $\mathbf{p}^0$ contained values calculated using Formula (8).

$$p^0 = \frac{Capacity}{\sum_{i=1}^{n} weight_i} \tag{8}$$

The initial probability distribution was assumed to be uncorrelated. Therefore, matrix $\mathbf{C}^0$ was a diagonal matrix with the variances computed according to Formula (9) located on the diagonal.

$$Var^0 = p^0 \cdot (1 - p^0) \tag{9}$$

*Test Environment*

Since CMA-PBIL requires the generation of binary vectors with a given correlation, it was important to select an algorithm that provided this. There are several types of algorithms for this [13–15]. One of them was proposed by Demitras and is a modification of the Emrich and Piedemont algorithm. An implementation of the algorithm from the MultiOrd package [16] in R was adopted in the experiments. The algorithm uses a correlation matrix instead of a covariance matrix, on which a constraint (10) is imposed, where $\psi_i^t$ is described by Formula (11).

$$max(-\psi_i^t \psi_j^t, \frac{-1}{\psi_i^t \psi_j^t}) \leqslant Cor^t(i,j) \leqslant min(\frac{\psi_i^t}{\psi_j^t}, \frac{\psi_j^t}{\psi_i^t}) \tag{10}$$

$$\psi_i^t = \sqrt{\frac{p^t(x_i)}{(1 - p^t(x_i))}} \tag{11}$$

In the PBIL algorithm, three parameters could be specified: $M$—the number of solutions generated; $N$—the number of solutions taken into account when updating the distribution parameters; $a$—the learning rate. In addition, the penalty factor $P$ was introduced for the knapsack problem. $\epsilon$ and $max_{iter}$ parameters determined the stop condition of the algorithms. The value of the factor prevented solutions exceeding the knapsack constraint from being selected. Parameter settings are listed in Table 2.

**Table 2.** Algorithm parameters.

| Parameter | Value |
|:---:|:---:|
| $M$ | 100 |
| $N$ | 20 |
| $a$ | $\{0.1, 0.5\}$ |
| $P$ | 1000 |
| $\epsilon$ | 0.1% |
| $max_{iter}$ | 1000 |

## 4. Results and Discussion

Each experiment was conducted 50 times. Figures 1–6 show the changes in the average of the maximum obtained values of the objective function so far during the subsequent evaluation. The best solutions found so far were captured after 100, 200, 500, 1000, 2000, 5000, 10,000, 20,000, 50,000 and 100,000, performing the objective function calculation and average for each test case after all experiments. To compare the results with the global optimum, the problem was also solved using Mixed-Integer Programming (MIP).

It was apparent from the presented graphs that both algorithms performed very similarly. For most test cases carried out, higher achieved values for the CMA-PBIL algorithm could be observed in the initial phases.

In addition, Figures 1, 3 and 5 noticeably show that, for knapsacks of capacity equal $2 \cdot v$ for both tested algorithms at a learning rate of $a = 0.5$, the computation terminated prematurely by converging to a local extreme.

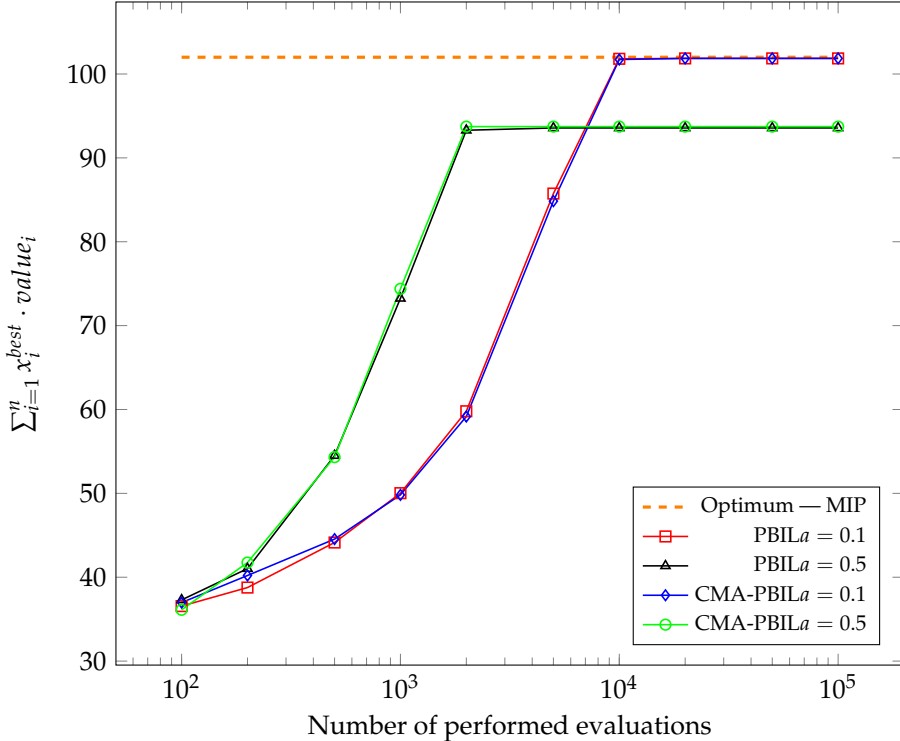

**Figure 1.** The average of the maximal values of the objective function calculated so far, test case no 1.

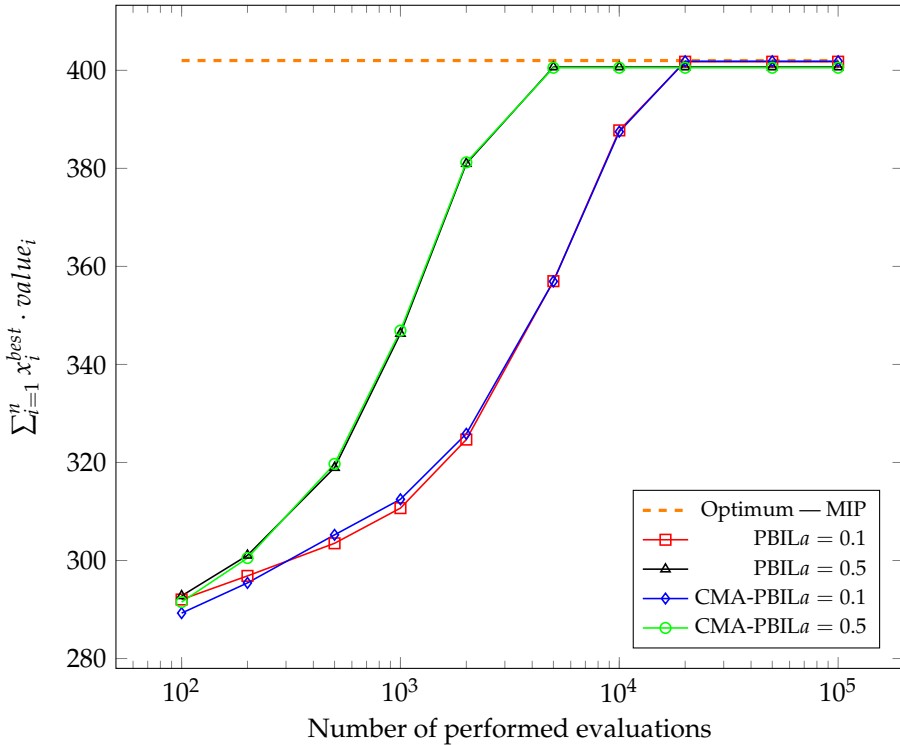

**Figure 2.** The average of the maximal values of the objective function calculated so far, test case no 2.

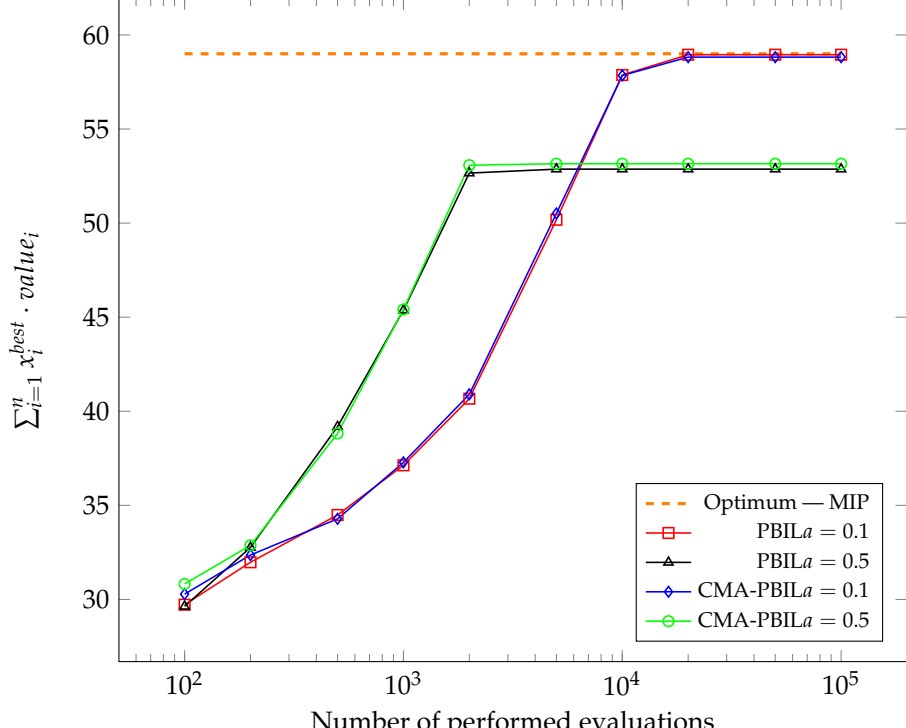

**Figure 3.** The average of the maximal values of the objective function calculated so far, test case no 3.

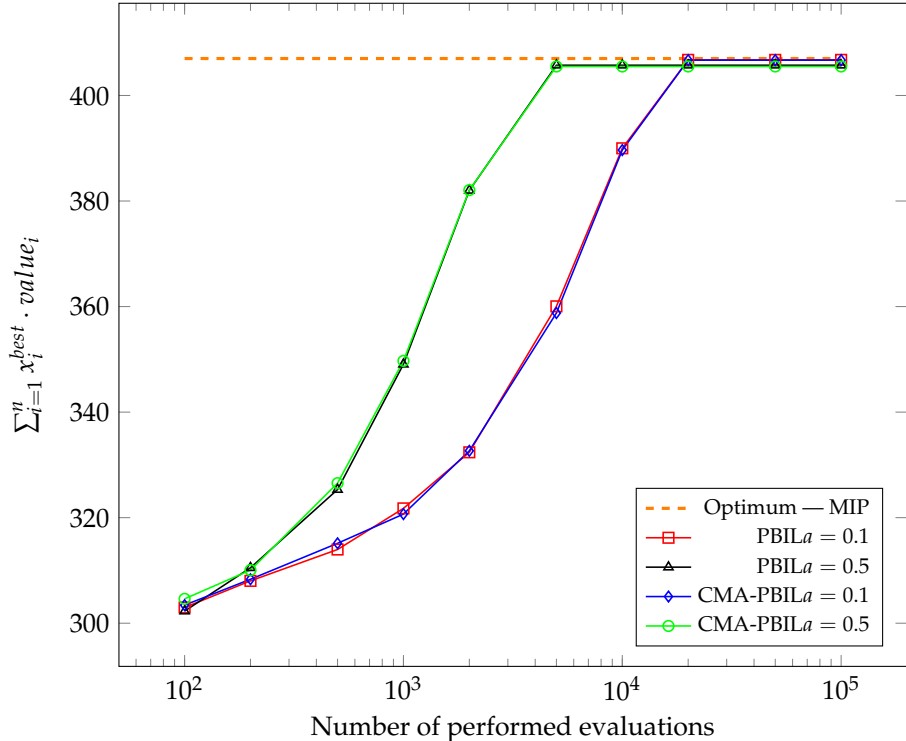

**Figure 4.** The average of the maximal values of the objective function calculated so far, test case no 4.

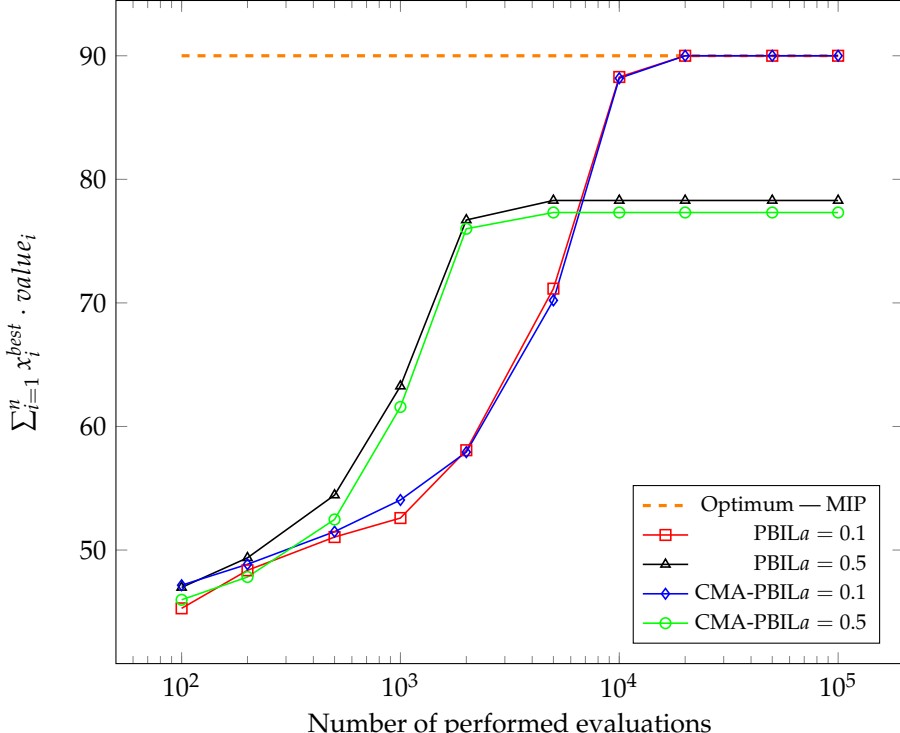

**Figure 5.** The average of the maximal values of the objective function calculated so far, test case no 5.

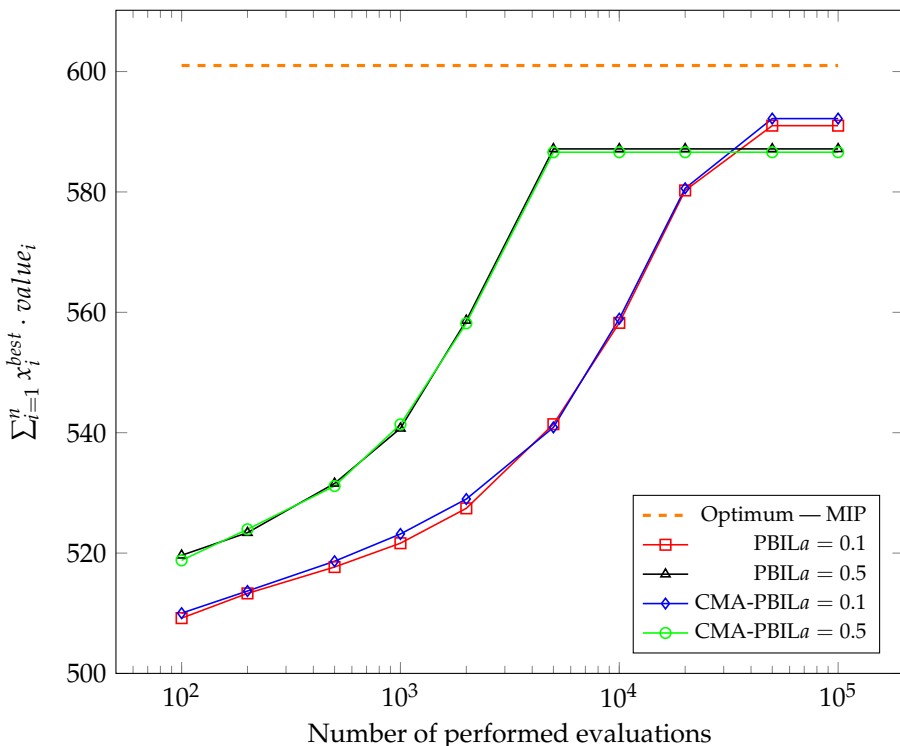

**Figure 6.** The average of the maximal values of the objective function calculated so far, test case no 6.

Tables [3] and [4] show the mean values and standard deviations of the objective function and the number of its evaluation in each test case. The last column in each table contains the *p*-value of the Wilcoxon–Mann–Whitney test, which was selected since the results did not come from a normal distribution. The most significant difference favouring the CMA-PBIL algorithm could be observed for test case number five, a knapsack problem with a strong correlation between weights and item values and a small capacity and a learning rate of $a = 0.1$. For both algorithms, the final average value of the test case's objective function was identical, while the CMA-PBIL algorithm required about 1700 less evaluations of the objective function. For no other test case did the PBIL algorithm perform noticeably better—the differences, sometimes in favour of the reference version and sometimes in favour of the modified version, were marginal.

The modifications that were proposed have in no case resulted in an evident degradation of the results. On the contrary, it was possible to identify a case for which the consideration of the covariance matrix in the probability distribution yielded an improvement in the convergence of the $\mathbf{p}^t$ vector.

The covariance matrix was intended to benefit only in specific test cases, particularly where there were strongly competing solutions. Such instances introduce a high correlation between the variables themselves, as in test problem number five. If one looked closely at how items were generated there, it would be clear that many items with identical parameters were generated. When looking at the value of the objective function, it was irrelevant which of the identical objects was included in the solution and, moreover, the constraint may cause that both at the same time should not be packed. Taking correlation into account allowed for a quicker separation of solutions giving the same result.

**Table 3.** Comparison of mean values and standard deviations of the objective function for each test case.

| No. | CMA-PBIL | | PBIL | | MIP | *p*-Value |
|---|---|---|---|---|---|---|
| | **Mean** | **Std** | **Mean** | **Std** | | |
| | | | **(a)** $a = 0.1$ | | | |
| 1 | 101.86 | 0.35 | 101.86 | 0.35 | 102 | 0.972 |
| 2 | 401.85 | 0.36 | 401.76 | 0.43 | 402 | 0.121 |
| 3 | 58.82 | 0.45 | 58.95 | 0.22 | 59 | 0.136 |
| 4 | 406.66 | 0.84 | 406.75 | 0.8 | 407 | 0.563 |
| 5 | 90.00 | 0 | 90.00 | 0 | 90 | – |
| 6 | 592.18 | 39.52 | 591.01 | 39.65 | 601 | 0.011 |
| | | | **(b)** $a = 0.5$ | | | |
| 1 | 93.73 | 4.72 | 93.57 | 4.81 | 102 | 0.719 |
| 2 | 400.53 | 1.33 | 400.66 | 2.54 | 402 | 0.052 |
| 3 | 53.15 | 2.77 | 52.87 | 3.03 | 59 | 0.634 |
| 4 | 405.49 | 1.43 | 405.73 | 1.31 | 407 | 0.261 |
| 5 | 77.32 | 5.25 | 78.29 | 4.83 | 90 | 0.363 |
| 6 | 586.60 | 4.06 | 587.15 | 3.54 | 601 | 0.6 |

**Table 4.** Comparison of the average number of objective function evaluation and its standard deviation for each test case.

| No. | CMA-PBIL | | PBIL | | *p*-Value |
|---|---|---|---|---|---|
| | **Mean** | **Std** | **Mean** | **Std** | |
| | | | **(a)** $a = 0.1$ | | |
| 1 | 20,052 | 1456.2 | 19,528 | 1320.1 | 0.076 |
| 2 | 33,822 | 2409.9 | 33,503 | 2508.9 | 0.310 |
| 3 | 23,908 | 3428.9 | 23,641 | 2795.3 | 0.738 |
| 4 | 35,754 | 2649 | 36,210 | 3001.4 | 0.305 |
| 5 | 31,931 | 4201.3 | 33,633 | 3585.2 | 0.052 |
| 6 | 59,978 | 7979 | 59,957 | 8514.4 | 0.905 |
| | | | **(b)** $a = 0.5$ | | |
| 1 | 3600 | 415.4 | 3746 | 557.2 | 0.380 |
| 2 | 6116 | 593.1 | 6089 | 598.57 | 0.861 |
| 3 | 3705 | 447.8 | 3911 | 583.9 | 0.112 |
| 4 | 6390 | 552.1 | 6373 | 533.2 | 0.732 |
| 5 | 4550 | 535.1 | 4666 | 545.9 | 0.326 |
| 6 | 8618 | 456. | 8505 | 473.21 | 0.122 |

*Potential Applications*

Nowadays, optimization issues arise in almost every area of science, engineering and economics. Model-based optimization using probabilistic modelling of the search space is a potential area for research of evolutionary algorithms. The PBIL is one of the algorithms that has been extensively applied to many optimization problems, both in the continuous and discrete domains. The CMA-PBIL algorithm proposed in the paper gave satisfactory results in the knapsack problem. Therefore, we will apply it to a similar problem concerning simulation and analysis methods of logistics networks for the postal operator. It will be our continuation and future work of application of the CMA-PBIL algorithm. The postal problem is a kind of composite of two problems. Firstly, it is a logistics problem, which considers how to determine the optimal set of routes with a fleet of vehicles to meet customer demands at the lowest possible cost. It is a generalization of the well-known travelling salesman problem. In addition, each car has to carry many items

that can be distributed among the machines in very many ways. Secondly, the question is the same as in the knapsack problem—how to allocate the goods so that their total weight and size do not exceed the vehicle's capacity and the total value transport is as high as possible.

Another problem in the discrete solution space where CMA-PBIL will find a potential application is the problem of optimizing node resources in a Dense Wavelength Division Multiplexing (DWDM) optical network, described in [17]. The main objective of the optimization is to minimize capital expenditure, which includes the costs of optical node resources, such as transponders and amplifiers used in a new generation of optical networks. A model, taking into account the physical phenomena in the optical network, was proposed.

The problems mentioned above were, in our view, appropriate where the CMA-PBIL algorithm would be competitive with the methods used there. Preliminary studies supported our thesis.

## 5. Conclusions

This paper investigated the impact of modifying the PBIL algorithm to include dependencies between variables on convergence and performance. The idea behind the CMA-PBIL algorithm was to introduce a covariance matrix to describe the probability distribution that represents the populations, which was inspired by the CMA-ES algorithm. To the best of the authors' knowledge, no studies attempted to add a covariance matrix to the binary distributions in the PBIL algorithm.

The focus of the experiments investigating CMA-PBIL was to determine how the modification affected the quality of the resulting solutions. The intention was that the CMA-PBIL algorithm should perform no worse than the standard PBIL and find a solution faster in specific cases. Both algorithms were tested on different types of knapsack problems, differing in the level of correlation between object attributes and knapsack capacity. The results confirmed the assumption, and for the highly correlated case having equivalent solutions, the CMA-PBIL algorithm completed it with a reduced number of calculation of the objective function, reaching the optimal result. While CMA-PBIL only benefited a specific class of problems, it is notable that it did not visibly degrade the result if the problem did not fit into this class.

In practical applications, some problems could also reach the same state in many equivalents and equally costly ways. Therefore, it is worthwhile to consider the CMA-PBIL algorithm for their solving since, similarly, as shown on the knapsack problem, if there are indications that there are strong correlations between the decision variables, a faster convergence can be achieved.

**Author Contributions:** Conceptualization, J.A. and S.K.; methodology, J.A., S.K., M.K. and A.P.; software, M.K. and A.P.; validation, S.K. and J.A.; formal analysis, S.K. and J.A.; investigation, J.A., S.K., A.P. and M.K.; resources, J.A. and S.K.; data curation, A.P. and M.K.; writing—original draft preparation, M.K., A.P. and S.K.; writing—review and editing, S.K. and J.A.; visualization, A.P. and M.K.; All authors have read and agreed to the published version of the manuscript.

**Funding:** This research was funded by the National Center for Research and Development under grant POIR.04.01.04-00-0054/17-00). LAS Project under the title *Simulation and analysis methods of logistics networks for postal operators*. Funding of The National Centre for Research and Development in Poland.

**Institutional Review Board Statement:** Not applicable.

**Informed Consent Statement:** Not applicable.

**Data Availability Statement:** Not applicable

**Acknowledgments:** The authors would like to thank the anonymous reviewers for their insightful, relevant and valuable comments.

**Conflicts of Interest:** The authors declare no conflict of interest.

## Abbreviations

The following abbreviations are used in this manuscript:

| | |
|---|---|
| EA | Evolutionary Algorithm |
| EDA | Estimation of Distribution Algorithm |
| PBIL | Population-Based Incremental Learning |
| ES | Evolution Strategy |
| CMA-ES | Covariance Matrix Adaptation Evolution Strategy |
| CMA-PBIL | Covariance Matrix Adaptation Population-Based Incremental Learning |
| DWDM | Dense Wavelength Division Multiplexing |
| MIP | Mixed-Integer Programming |

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
