# Peer review of "A Modification of the PBIL Algorithm Inspired by the CMA-ES Algorithm in Discrete Knapsack Problem"

_applsci, doi:10.3390/app11199136_

Round 1
Reviewer 1 Report
The paper is not very original. It smartly adapts the results from previous papers to another setting. Still, the results seems sound from the experimental evaluation and the problem is attracting a lot of recent interest.
But, totally thinking, this is not enough for the publication.
I think you should test your algorithm against CMA-ES as well (seems fair since it's part of your algorithm).
You should explain why (since it seems strange to me) in Figures 1,3,5 the algorithms reached a local maximum.
You should compare your algorithm to the optimal (you can find many algorithms in the literature to find the optimal for the knapsack problem).
With this it may also be possible to understand how far your solutions are from the optimal.
The figures should be changed to make them more understandable. For example by using line chart symbols. Also change the colors of the PBIL algorithms (it may be my computer but it is hard to distinguish the two colors).
Author Response
Dear Sir,
We wish to thank you for your careful reading of the manuscript and for a set of helpful comments. We hope that after the revision the manuscript will find the approval of the editorial board. The changes to the manuscript are listed below. In the text, we use yellow highlighted text to ease the tracking of the changes. The details of these changes we provide in the reply to reviewers’ comments are in the attached file.
Kind regards,
Stanislaw Kozdrowski

Reviewer 2 Report
The aim of the paper is good, the realisation is not perfect, but I think that it is possible to increase the paper quality.
The English-level need to be corrected by Native speaker.
Formally:
- in eq2 it is not clear what is 'x' (it is maybe hidden somewhere in previous text)
- Fig1-Fig6 - tehre is no difference between PBIL variants line-colours.
Practically:
- The new method has to be described more clearly (try to use one pseudocode, etc) in a more compact form.
- Authors have to use more than one problem to compare the new method (even the problem is well-known).
- Authors (from my point of view) have to employ some different methods - known for this topic of problems (to indicate the quality of proposed method).
- There is no statistical comparison (using the standard tests) - it is necessary.
Author Response

(The authors gave the same response as above.)

Round 2
Reviewer 1 Report
I have read the letter from the authors and have seen their changes to the article.
However, I remain of my opinion that this is not sufficient for publication.
The authors smartly create a new algorithm that combines two existing approaches, but there are no significant improvements in the result over the previous algorithm known in the literature.
I would like to add that the authors write (in the conclusions): "The intention is that the CMA-PBIL algorithm should perform no worse than standard PBIL and find a solution faster in specific cases"; but I do not see any analysis (mathematical or empirical) of the time consumed by algorithms.
Author Response
Dear Sir,
We wish to thank you for your careful reading of the manuscript and a set of helpful comments. We hope that after the revision, the manuscript will find the approval of the editorial board. The changes to the manuscript are listed below. In the text, we use yellow highlighted text to ease the tracking of the changes. The details of these changes we provide in reply to reviewers’ comments. Please find the pdf attached file.
Kind regards,
Stanislaw Kozdrowski

Reviewer 2 Report
Dear authors, thank toy for a fast update of the text. Nevertheless, the 'statistical comparison' means using some proper statistical method/test to distinguish the real and the random differences between results. I still insist on the regular statistical comparison of the results, if the paper has to be printed in IF journal.
Author Response

(The authors gave the same response as above.)

Round 3
Reviewer 1 Report
I can see that authors have added more details etc. to make it more reasonable to get accepted as a Journal paper, not a bad work in such a short time I have to say.